

# Integrative microRNA-gene expression network analysis in genetic hypercalciuric stone-forming rat kidney

Yuchao Lu, Baolong Qin, Henglong Hu, Jiaqiao Zhang, Yufeng Wang, Qing Wang and Shaogang Wang

Institute and Department of Urology, Tongji Hospital, Tongji Medical College, Huazhong University of Science and Technology, Wuhan, PR China

## ABSTRACT

**Background.** MicroRNAs (miRNAs) influence a variety of biological functions by regulating gene expression post-transcriptionally. Aberrant miRNA expression has been associated with many human diseases. Urolithiasis is a common disease, and idiopathic hypercalciuria (IH) is an important risk factor for calcium urolithiasis. However, miRNA expression patterns and their biological functions in urolithiasis remain unknown.

**Methods and Results.** A multi-step approach combining microarray miRNA and mRNA expression profile and bioinformatics analysis was adopted to analyze dysregulated miRNAs and genes in genetic hypercalciuric stone-forming (GHS) rat kidneys, using normal Sprague-Dawley (SD) rats as controls. We identified 2418 mRNAs and 19 miRNAs as significantly differentially expressed, over 700 gene ontology (GO) terms and 83 KEGG pathways that were significantly enriched in GHS rats. In addition, we constructed an miRNA-gene network that suggested that rno-miR-674-5p, rno-miR-672-5p, rno-miR-138-5p and rno-miR-21-3p may play important roles in the regulatory network. Furthermore, signal-net analysis suggested that NF-kappa B likely plays a crucial role in hypercalciuria urolithiasis.

**Conclusions.** This study presents a global view of mRNA and miRNA expression in GHS rat kidneys, and suggests that miRNAs may be important in the regulation of hypercalciuria. The data provide valuable insights for future research, which should aim at validating the role of the genes featured here in the pathophysiology of hypercalciuria.

Corresponding author
Shaogang Wang,
sgwangtjm@163.com,
sgwangtjm@126.com

## INTRODUCTION

Kidney stones are commonly found in children and adults (*Coe, Evan & Worcester, 2005*), and can becaused by multiple factors. Idiopathic hypercalciuria (IH) is an important risk factor for calcium urolithiasis (*Worcester et al., 2013*; *Yoon et al., 2013*). Patients with IH have normal serum $Ca^{2+}$, and increased urinary calcium excretion. But the pathophysiological process and molecular mechanism of IH are still unclear. The genetic hypercalciuric stone-forming (GHS) rat, has many pathophysiological characteristics identical to that of IH patients, such as normal serum $Ca^{2+}$, hypercalciuria, elevated intestinal $Ca^{2+}$

resorption and a tendency to lose calcium from the bone (*Frick et al., 2013*; *Frick, Krieger & Bushinsky, 2015*), which is regarded as an ideal animal model of calcium urolithiasis.

MicroRNAs (miRNAs) are a group of small, non-coding RNAs that regulate protein-coding gene function at the post-transcriptional level by binding to complementary sites on target mRNAs in the 3′UTR (*Ambros, 2004*). Meanwhile, miRNAs have been shown to regulate a wide range of biological processes including cell growth, metabolism, differentiation, proliferation and apoptosis, which have important implications in diseases processes (*Ambros, 2001*). Dysregulation of miRNAs has been associated with many human diseases. However, miRNA expression patterns and their biological functions in urolithiasis remain unknown. Understanding the relevance of miRNA and mRNA expression patterns in GHS rat kidneys is important to better elucidate the relationship between pathophysiological process and genes.

In this study we therefore conducted mRNA and miRNA expression profiling of three pairs of GHS and normal Sprague-Dawley (SD) rat kidneys. A subset of differentially expressed genes was validated by qPCR in 12 pairs of kidneys. Bioinformatic analysis was further performed to construct an integrative regulatory network of altered miRNA-mRNA transcriptsin GHS rat kidney.

## MATERIALS AND METHODS

### Animals
The colony of GHS rats were created by selective breeding of male and female Sprague-Dawley (SD) rats with the highest 24-hour urine calcium excretion as previously described (*He et al., 2015*). By the 28th generation, GHS rats stably excreted significantly higher levels of urine calcium than wild-type normal SD rats. Normal SD rats were purchased from the Experimental Animal Center, Tongji Medical College, Huazhong University of Science and Technology, China. Twelve GHS rats with a body weight of 250–280 g were used for this study. A total of 12 normal SD rats were matched with GHS rats with respect to body weight and age, and served as control rats. All rats were fed 13 g/day of a normal calcium diet (1.2% calcium, 0.65% phosphorus, 0.43% chloride, 0.4% sodium, and 0.24% magnesium per gram of food). All animal procedures were approved by the Ethical Committee of Tongji Hospital, Tongji Medical College, Huazhong University of Science and Technology (No. TJ-A20141211). All surgeries were performed under sodium pentobarbital anesthesia.

### Urine and serum calcium and phosphorus determination
Two successive 24-hour urine samples were collected before rats were killed, then blood samples were taken after rats were killed. Urine calcium, serum calcium, and phosphorus were measured using an Abbott Aeroset AutoAnalyzer (Abbott Diagnostics, Chicago, IL, USA).

### RNA extraction
Total RNA was extracted from kidney tissue using RNeasy Fibrous Tissue kit (Qiagen, Dusseldorf, Germany) according to the manufacturer's protocol. RNA purity and concentration were assessed by NanoDrop ND-2000 spectrophotometer (Thermo Fisher

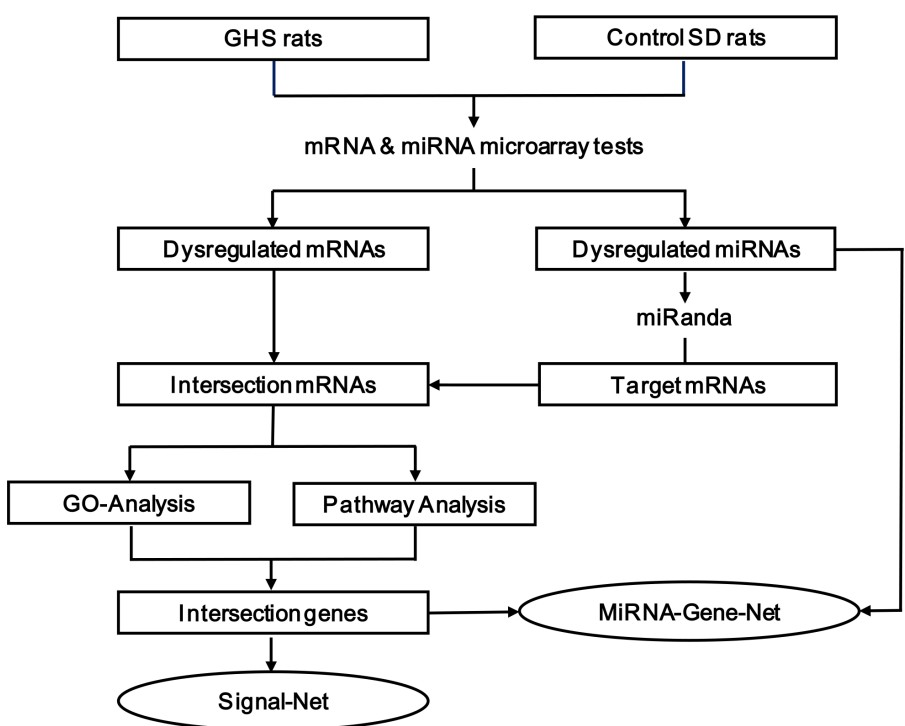

**Figure 1** The multi-step strategy used in this study.

Scientific Inc., Waltham, MA, USA) and electrophoresis of RNA on agarose gel containing formaldehyde was used to evaluate the integrity of RNA.

## Microarray

Three GHS rats and three SD rats were randomly selected for microarray analysis. The Affymetrix GeneChip miRNA 4.0 Array and Affymetrix Gene 1.0 Array for rats were used to compare miRNA and mRNA expression profiles, respectively, in GHS and control rat kidneys. Microarray analysis was performed by GMINIX Informatics Ltd. Co, Shanghai, China. The data have been deposited in the NCBI Gene Expression Omnibus and are accessible through GEO Series accession number GSE75543.

## Strategy

As shown in Fig. 1, we used a multi-step strategy to identify genes dysregulated in GHS rats relative to the control group. First, significantly differentially expressed mRNA and miRNA were identified using a random variance model (RVM) corrective analysis of variance (ANOVA) (*Wright & Simon, 2003*). Second, the miRanda database was used to predict putative miRNA targets, and the overlap between these target genes and differentially expressed mRNAs was established. Third, these intersecting genes were classified according to their biological functions using the Gene Ontology System. Similarly, pathway analysis was used to identify affected KEGG pathways. Fourth, genes that were present in both the enriched GO terms and significant KEGG pathways were used to construct a miRNA-gene network (*Joung et al., 2007*; *Shalgi et al., 2007*) and signal-net (*Spirin & Mirny, 2003*; *Zhang & Wiemann, 2009*). The center of the network is represented by a degree, which indicates

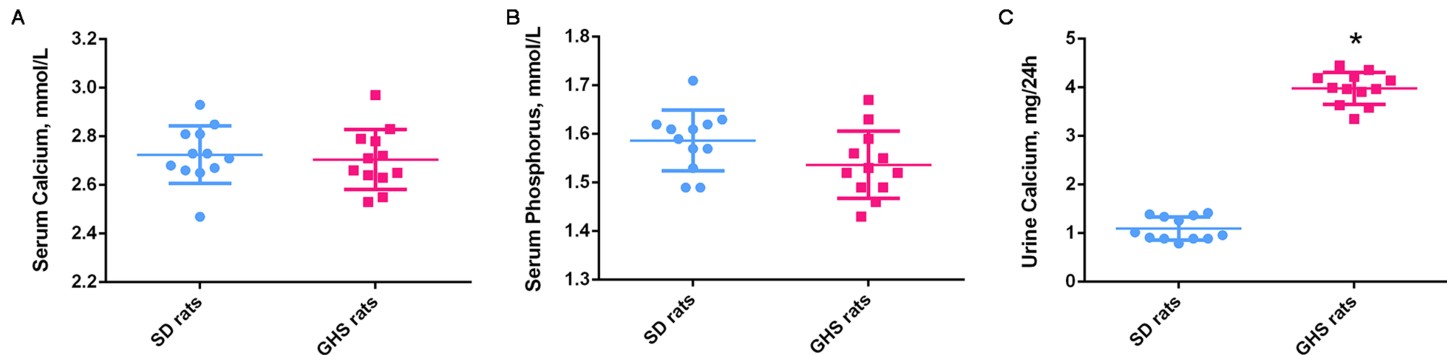

**Figure 2  Serum calcium, phosphorus and urine calcium levels of GHS and normal SD rats.** The serum calcium (A) and serum phosphorus (B) levels were not significantly altered in GHS compared with SD rats. The 24-h urine calcium (C) level in GHS rats was significantly increased. Symbol (*) indicates significant difference from control ($P < 0.05$).

the predicted interaction of a given miRNA with its target genes. All of the data mentioned above were analyzed by GCBI working platform (GMINIX Informatics Ltd. Co, Shanghai, P. R. China), and the principles and methodologies of data analysis were described in File S1. Finally, a subset of the predicted miRNA-mRNA pairs was selected for validation by qRT-PCR in an extended cohort of kidney tissues.

### Real-time RT-PCR

For mRNA, total RNA was extracted and 500 ng of RNA was used for cDNA synthesis using the Takara reverse transcription kit (Takara, Dalian, China) according to the manufacturer's instructions. PCR was conducted using SYBR Premix Ex Taq (Takara, Dalian, China) according to the manufacturer's instructions on an Mx3000P system (Agilent Stratagene, Santa Clara, CA, USA). The primers were chemically synthesized by Tsingke, Wuhan, China and are listed in Table S1. The All-in-One™ miRNA qRT-PCR Detection Kit (GeneCopoeia, Guangzhou, China) was used for both cDNA synthesis and quantitative detection using miRNA specific primers (GeneCopoeia, Guangzhou, China). GAPDH and U6 were used as internal controls to determine the relative expression of target mRNA and miRNA. All reactions were performed in triplicate.

### Statistical analysis

Continuous variables were expressed as means ± standard deviation. For each triplicate of microarray data, the geometric mean was used. Student's $t$-test was applied for comparisons between two groups, and ANOVA for comparisons between more than two groups. A $P$-value of <0.05 was considered statistically significant.

## RESULTS

### Serum calcium and phosphorus levels, and urine calcium excretion

Serum calcium and phosphorus levels were not significantly altered in GHS compared with SD control rats (Figs. 2A and 2B). GHS rats did, however, excrete significantly more urine calcium (mg/24 h) than SD control rats (Fig. 2C).

**Table 1  19 dysregulated miRNAs.** The significantly dysregulated miRNAs between GHS and SD rats.

| miRNA | *P*-value | Fold-change | Style |
|---|---|---|---|
| rno-miR-184 | 0.00069 | 29.41 | up |
| rno-miR-21-3p | 0.001198 | 4.17 | up |
| rno-miR-672-5p | 0.047372 | 2.70 | up |
| rno-miR-6324 | 0.018993 | 2.50 | up |
| rno-miR-154-5p | 0.028899 | 2.13 | up |
| rno-miR-770-3p | 0.040784 | 1.64 | up |
| rno-miR-674-5p | 0.003039 | 1.59 | up |
| rno-miR-376a-3p | 0.02666 | 1.52 | up |
| rno-miR-99b-3p | 0.049668 | 1.25 | up |
| rno-miR-146a-5p | 0.040208 | 1.18 | up |
| rno-miR-203b-3p | 0.018288 | 0.65 | down |
| rno-miR-20b-3p | 0.032176 | 0.64 | down |
| rno-miR-206-3p | 0.006578 | 0.45 | down |
| rno-miR-196c-3p | 0.028658 | 0.38 | down |
| rno-miR-138-5p | 0.018426 | 0.32 | down |
| rno-miR-203a-3p | 0.03408 | 0.30 | down |
| rno-miR-201-3p | 0.03048 | 0.25 | down |
| rno-miR-138-1-3p | 0.00485 | 0.11 | down |
| rno-miR-484 | 0.004176 | 0.08 | down |

## miRNA and mRNA expression profiles in GHS ratkidney

The miRNA expression profiles of three GHS rats' kidney tissues and the control rats were determined using miRNA microarray analysis. A total of 19 miRNAs were significantly differentially expressed between the two groups ($p < 0.05$), of which 10 and nine miRNAs were up- or downregulated, respectively, in GHS vs. control rats (Table 1 and Table S2); of these rno-miR-184, rno-miR-21-3p and rno-miR-672-5p had the largest positive fold changes, while rno-miR-484, rno-miR-138-1-3p and rno-miR-201-3p had the largest negative fold changes. The expression levels of these 19 miRNAs are illustrated by the heatmap in Fig. S1.

Microarray-based mRNA expression analysis was also conducted and 2,418 genes were identified as significantly differentially expressed in GHS rats ($P < 0.05$) including 1,057 upregulated genes and 1,361 downregulated genes (Table S3).

## miRNA target gene prediction

Target mRNAs for differentially expressed miRNAs were predicted using the miRanda database. Since miRNAs negatively regulate gene expression, upregulated miRNAs result in downregulated target mRNAs, and vice versa. A total of 29,164 miRNA-mRNA pairs (based on 19 dysregulated miRNAs and their 10,521 target mRNAs) were predicted (Table S4).

## Integrated analysis of dysregulated miRNAs and mRNAs

The set of intersecting mRNAs between the predicted target mRNAs and differentially expressed mRNAs were selected (Table S5) and those that were negatively correlated with

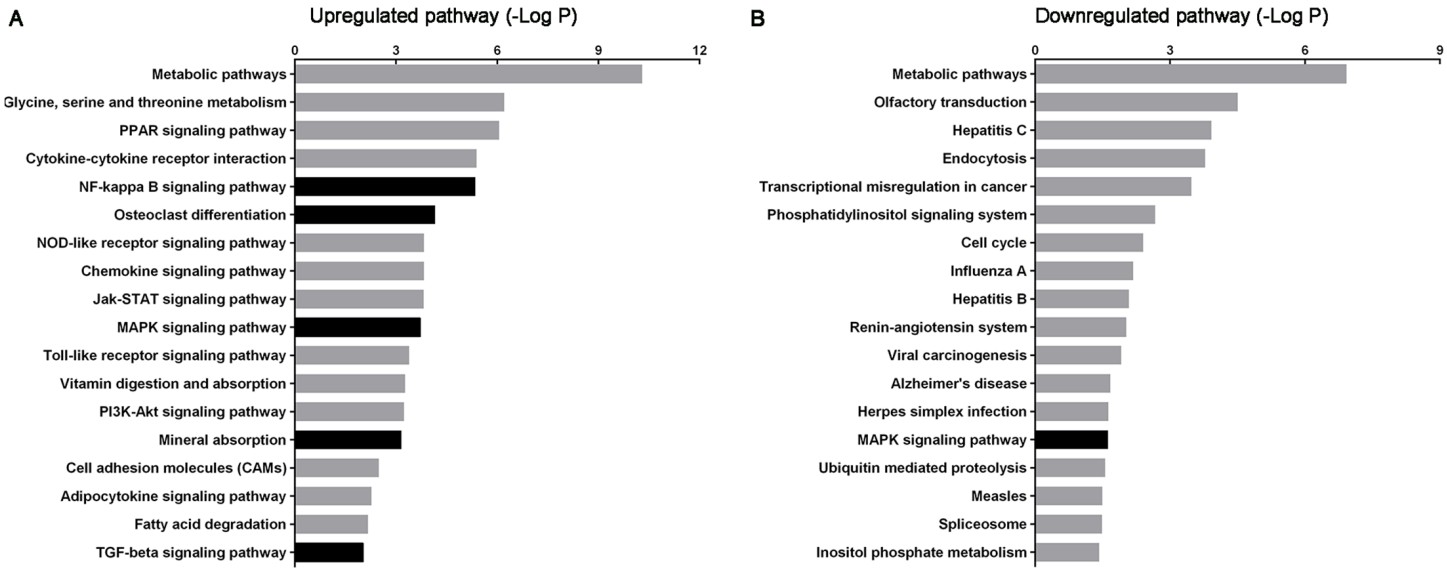

**Figure 3   Histogram of signaling pathways based on the intersecting genes from predicted target mRNAs and differentially expressed mRNAs.** (A) and (B) show significant upregulated and downregulated pathways, respectively. $X$-axis, negative logarithm of the $P$-value ($-$LgP); $Y$-axis, pathway. The higher the $-$LgP, the lower the $P$-value. The darker bars indicate pathways relate to urolithasis.

their predicted miRNA matches were used for downstream GOanalysis and KEGG pathway analysis.

There were 417 upregulated and 286 downregulated GO terms ($P < 0.05$), with the most significant GO terms including negative regulation of apoptotic process (GO:0043066), response to lipopolysaccharide (GO:0032496) and inflammatory response (GO:0006954). Table 2 shows the top 15 up- and downregulated GO terms, with further detail in Tables S6 and S7.

According to KEGG pathway analysis, 93 KEGG pathways were significantly enriched, of which 75 were upregulated and 18 downregulated at $P < 0.05$ (Tables S8 and S9). The most highly enriched pathways included the Pantothenate and CoA biosynthesis and synthesis and degradation of ketone bodies pathways. The top up- and downregulated pathways are shown in Fig. 3. The darker bars indicate pathways that reported to be related to urolithasis in previous studies.

Next, to illustrate the role of key miRNAs in the regulation of kidney mRNAs in GHS rats, a miRNA-mRNA-Network was built based on the subset of significantly differentially expressed mRNAs that were also members of significantly enriched GO terms and KEGG pathways (Table S10, Fig. 4). In total, 223 mRNAs and 19 miRNAs were included in the network, where box nodes represent miRNAs, and circular nodes represent mRNAs. The degree of connectivity, which represents the number of genes regulated by a given miRNA, is indicated by the size of the node with a higher degree of connectivity represented by larger nodes (Table S11). Rno-miR-674-5p, rno-miR-672-5p, rno-miR-138-5p, and rno-miR-21-3p had high degrees of connectivity and may play crucial roles in this regulatory network. Meanwhile, Sema5a, Lpin2, Gcnt1, Masp1, Olr1 and Traf3 were the most common miRNA targets. Table 3 presented a subset of significantly regulated miRNA-mRNA hybrids.

**Table 2 GO-dysregulated.** The top 15 up- and downregulated GO terms.

| GO-ID | GO-name | Enrichment | *P*-value |
|---|---|---|---|
| **Upregulated GOs by GO analysis** | | | |
| GO:0043066 | Negative regulation of apoptotic process | 5.274431709 | 5.53034E−12 |
| GO:0032496 | Response to lipopolysaccharide | 8.108054973 | 1.51933E−11 |
| GO:0006954 | Inflammatory response | 8.099023444 | 2.11218E−10 |
| GO:0007165 | Signal transduction | 5.49328365 | 2.53815E−10 |
| GO:0071347 | Cellular response to interleukin-1 | 16.88289078 | 1.09298E−08 |
| GO:0045944 | Positive regulation of transcription from RNA polymerase II promoter | 3.652056561 | 1.27896E−08 |
| GO:0014070 | Response to organic cyclic compound | 6.26379716 | 3.29652E−08 |
| GO:0008285 | Negative regulation of cell proliferation | 5.262199724 | 5.94266E−08 |
| GO:0033590 | Response to cobalamin | 90.04208417 | 6.01314E−08 |
| GO:0042493 | Response to drug | 4.287718294 | 6.34324E−08 |
| GO:0006468 | Protein phosphorylation | 4.846457788 | 8.88401E−08 |
| GO:0042542 | Response to hydrogen peroxide | 12.4673655 | 1.76279E−07 |
| GO:0009617 | Response to bacterium | 19.09983604 | 2.66E−07 |
| GO:0001889 | Liver development | 8.612721094 | 2.86336E−07 |
| GO:0001666 | Response to hypoxia | 5.58112092 | 4.40694E−07 |
| **Downregulated GOs by GO analysis** | | | |
| GO:0045893 | Positive regulation of transcription, DNA-dependent | 3.77523152 | 3.05206E−07 |
| GO:0045892 | Negative regulation of transcription, DNA-dependent | 4.062477396 | 3.32485E−07 |
| GO:0016568 | Chromatin modification | 13.26523231 | 5.79738E−07 |
| GO:0043065 | Positive regulation of apoptotic process | 5.038731654 | 6.66823E−07 |
| GO:0008150 | Biological process | 2.365931438 | 1.86723E−06 |
| GO:0016572 | Histone phosphorylation | 46.4283131 | 3.08593E−06 |
| GO:0045944 | Positive regulation of transcription from RNA polymerase II promoter | 2.954529015 | 4.94117E−06 |
| GO:0006355 | Regulation of transcription, DNA-dependent | 2.982729366 | 6.6796E−06 |
| GO:0003407 | Neural retina development | 36.11091019 | 1.08935E−05 |
| GO:0001525 | Angiogenesis | 5.879901494 | 1.29214E−05 |
| GO:0008284 | Positive regulation of cell proliferation | 3.559902873 | 3.26349E−05 |
| GO:0006950 | Response to stress | 5.762379285 | 4.3047E−05 |
| GO:0006915 | Apoptotic process | 3.631264712 | 4.68891E−05 |
| GO:0007165 | Signal transduction | 3.448256676 | 8.82113E−05 |
| GO:0043066 | Negative regulation of apoptotic process | 3.059606407 | 0.000134769 |

Moreover, to investigate the regulatory relationships between these genes and their potential role in hypercalciuria, we performed a signal-net analysis based on significantly regulated KEGG pathways (Fig. 5, Table S12). Signal-net analysis has shown that NF-kappa B signal pathway, including the members of NF-kappa B1, RelA and NF-kappa B2 et al., might play a core role in the gene regulatory network. And NF-kappa B1 and RelA had the highest degrees of connectivity of 19 and 18, respectively.

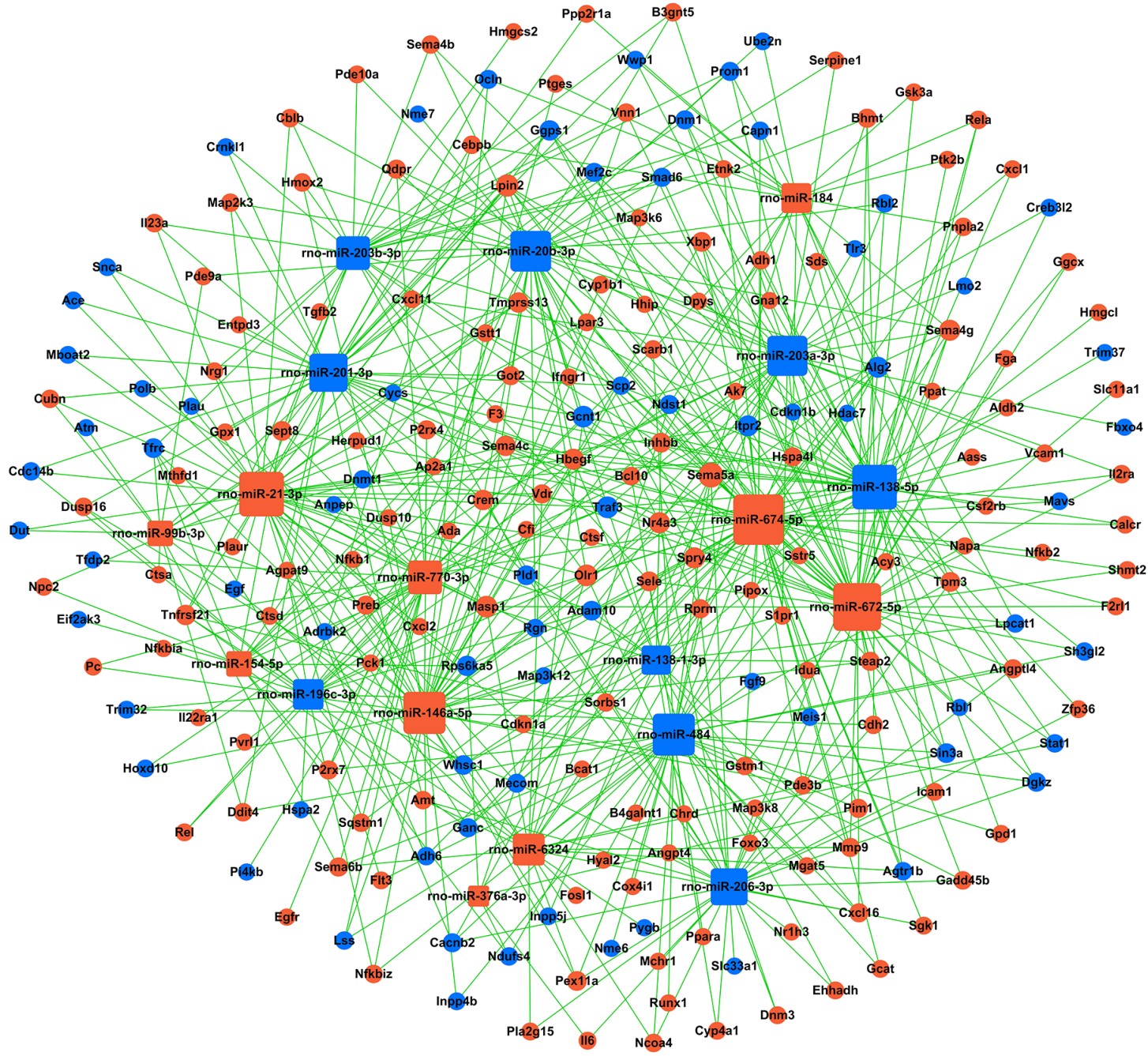

**Figure 4  miRNA-gene-network.** Box nodes represent miRNAs, circular nodes represent mRNAs. Blue represents downregulation, while represents upregulation. The higher the degree of connectivity of a gene, the larger the node within the network. In total, 223 mRNAs and 19 miRNAs were included in the network. Rno-miR-674-5p, rno-miR-672-5p, rno-miR-138-5p, and rno-miR-21-3p were found to have the highest degrees of connectivity.

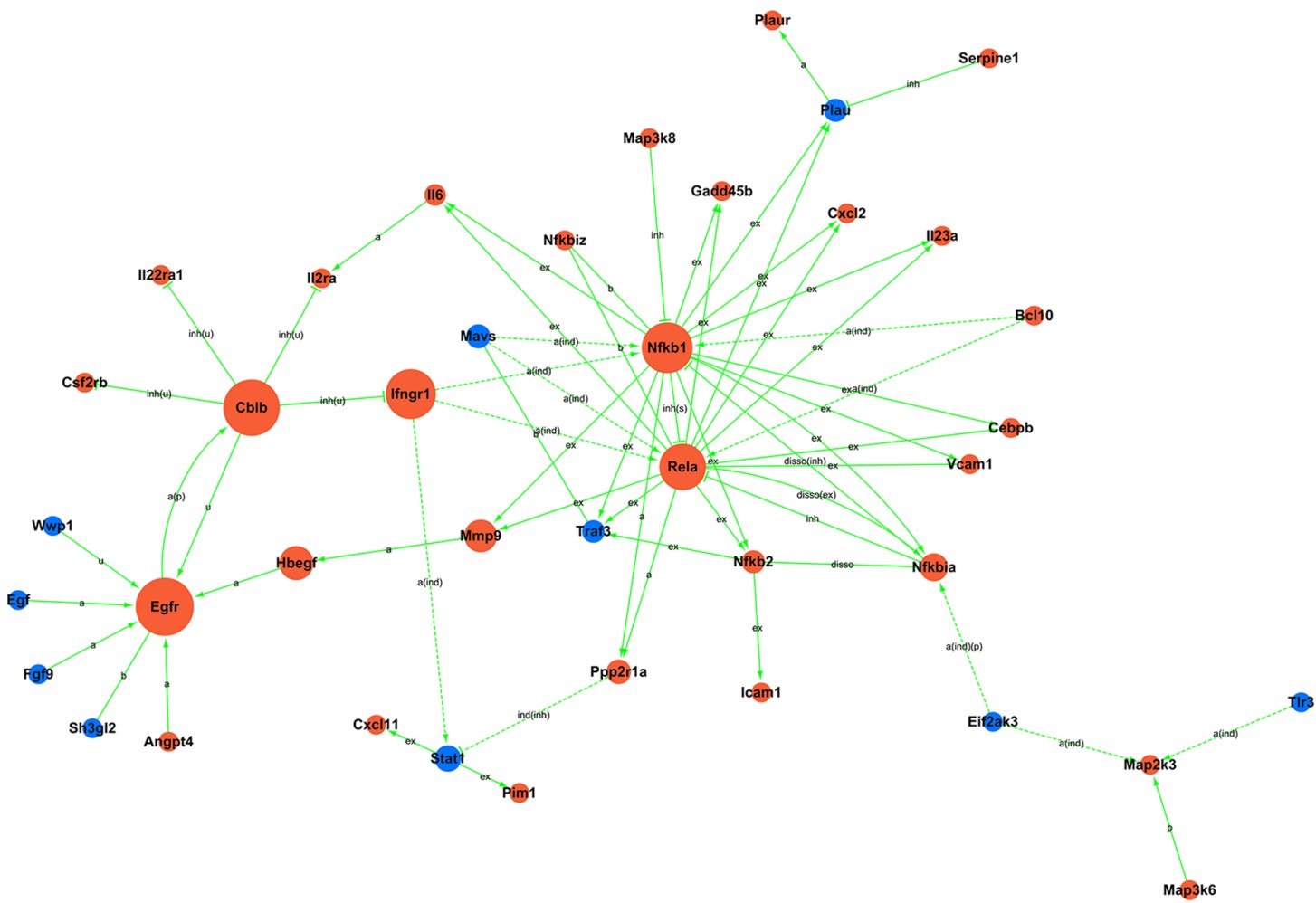

**Figure 5** **Signal-net.** Blue, downregulation; red, upregulation; a, activation; a(ind), activation (indirect effect); a(ind)(p), activation (indirect effect)(phosphorylation); a(p), activation (phosphorylation); b, binding/association; disso, dissociation; disso(ex), dissociation (expression); disso(inh), dissociation (inhibition); ex, expression; ind(inh), indirect effect (inhibition); inh, inhibition; inh(s), inhibition (state change); inh(u), inhibition (ubiquitination); p, phosphorylation; u, ubiquitination.

## Real-time quantitative PCR validation

To validate the reliability of our microarray-based results, 11 miRNAs and eight target mRNAs were selected for validation by qRT-PCR in 12 pairs of matched GHS/normal SD rat kidneys (Table S13). As demonstrated in Fig. 6A, rno-miR-184, rno-miR-484 and rno-miR-138-1-3p were the most significantly dysregulated miRNAs in GHS rats, and the results were comparable with our microarray data. Moreover, the expression levels of the seven mRNAs measured by qPCR (Gcnt1, Lpin2, Olr1, Sema5a, Nf$\kappa$ B1, Rela and VDR) coincided exactly with our microarray data except for CaSR (Fig. 6B).

These results demonstrate a strong consistency between the microarray- and qRT-PCR-based results.

Table 3 **miRNA-mRNA hybrid.** miRNAs and target mRNAs in GHS rat kidneys.

| miRNA | Target mRNAs |
|---|---|
| rno-miR-138-1-3p | Cxcl16, Cyp4a1, Dnm3, Ehhadh, Hbegf, Hspa4l, Inhbb, Masp1, Mmp9, Nr4a3, Rprm, Sema5a, Steap2 |
| rno-miR-138-5p | Aass, Adh1, Ak7, Bhmt, Calcr, Cdh2, Csf2rb, Cxcl1, F3, Ggcx, Gna12, Gpd1, Gsk3a, Gstt1, Hbegf, Hmgcl, Hspa4l, Icam1, Inhbb, Lpin2, Nr4a3, Pnpla2, Rela, Sds, Sema4b, Sema4c, Sema5a, Spry4, Tmprss13, Zfp36 |
| rno-miR-196c-3p | Agpat9, Cebpb, Dusp16, Hbegf, Inhbb, Masp1, Nfkbia, Nfkbiz, Olr1, Pvrl1, Rel, Sema4c, Sorbs1, Tnfrsf21 |
| rno-miR-201-3p | Amt, Bcl10, Crem, Ctsd, Cubn, Ddit4, Dusp16, Entpd3, Got2, Gpx1, Hbegf, Il23a, Lpin2, Map2k3, Masp1, Nr4a3, Ppp2r1a, Sema5a, Sept8, Tmprss13, Vdr |
| rno-miR-203a-3p | Angptl4, B3gnt5, Bcl10, Bhmt, Crem, Cxcl1, Cxcl11, Cyp1b1, Dpys, Hbegf, Il2ra, Inhbb, Lpin2, Nr4a3, Olr1, Qdpr, S1pr1, Sema4g, Sema5a, Serpine1, Sorbs1, Steap2, Tpm3, Vcam1, Xbp1 |
| rno-miR-206-3p | Amt, Angpt4, Angptl4, Cdh2, Crem, Cyp4a1, Dnm3, Ehhadh, Gadd45b, Icam1, Il6, Map3k8, Mchr1, Ncoa4, Nr1h3, Nr4a3, Olr1, Pex11a, Pla2g15, Sema4g, Sema5a, Sgk1, Spry4, Tpm3, Zfp36 |
| rno-miR-484 | Angptl4, B4galnt1, Bcl10, Chrd, Cox4i1, Cxcl16, Ddit4, Flt3, Foxo3, Gadd45b, Gcat, Gpd1, Hyal2, Mgat5, Ncoa4, Nfkbiz, P2rx7, Pex11a, Pim1, Ppara, Rprm, Runx1, S1pr1, Scarb1, Sele, Sema4g, Sema5a, Sema6b, Sorbs1, Tnfrsf21, Tpm3 |
| rno-miR-184 | Alg2, Dnm1, Lpcat1, Ndst1, Ocln, Prom1, Ube2n, Wwp1 |
| rno-miR-21-3p | Ace, Adam10, Adh6, Cacnb2, Cdc14b, Cycs, Dut, Egf, Ganc, Ggps1, Itpr2, Mboat2, Mecom, Ocln, Plau, Pld1, Rgn, Rps6ka5, Snca, Traf3, Whsc1 |
| rno-miR-672-5p | Agtr1b, Alg2, Creb3l2, Dnm1, Gcnt1, Ggps1, Itpr2, Lpcat1, Ndst1, Prom1, Rbl2, Scp2, Sh3gl2, Sin3a, Smad6, Stat1, Traf3 |
| rno-miR-674-5p | Adam10, Adh6, Agtr1b, Alg2, Capn1, Dgkz, Dnmt1, Ganc, Ggps1, Hdac7, Itpr2, Mavs, Mef2c, Ndufs4, Rbl1, Rbl2 |

## DISCUSSION

IH typically manifests as hypercalciuria with normal serum $Ca^{2+}$, increased intestinal $Ca^{2+}$ absorption, bone resorption and decreased bone mineral density in addition to decreased renal tubule $Ca^{2+}$ reabsorption (*Yoon et al., 2013*). Thus, IH is one of the most important risk factors for calcium urolithiasis. However, the pathogenesis of IH is not yet fully understood. GHS rats exhibit many features of human IH including increased intestinal $Ca^{2+}$ absorption, increased bone resorption, decreased renal tubule $Ca^{2+}$ reabsorption and high levels of vitamin D receptor (VDR) protein in $Ca^{2+}$-transporting organs (*Frick et al., 2013*; *Frick, Krieger & Bushinsky, 2015*). The research of *Tsuruoka, Bushinsky & Schwartz (1997)* strongly suggests that decreased tubular $Ca^{2+}$ reabsorption plays a key role in hypercalciuria. In general, GHS rats represent an ideal animal model for idiopathic hypercalciuria urolithiasis.

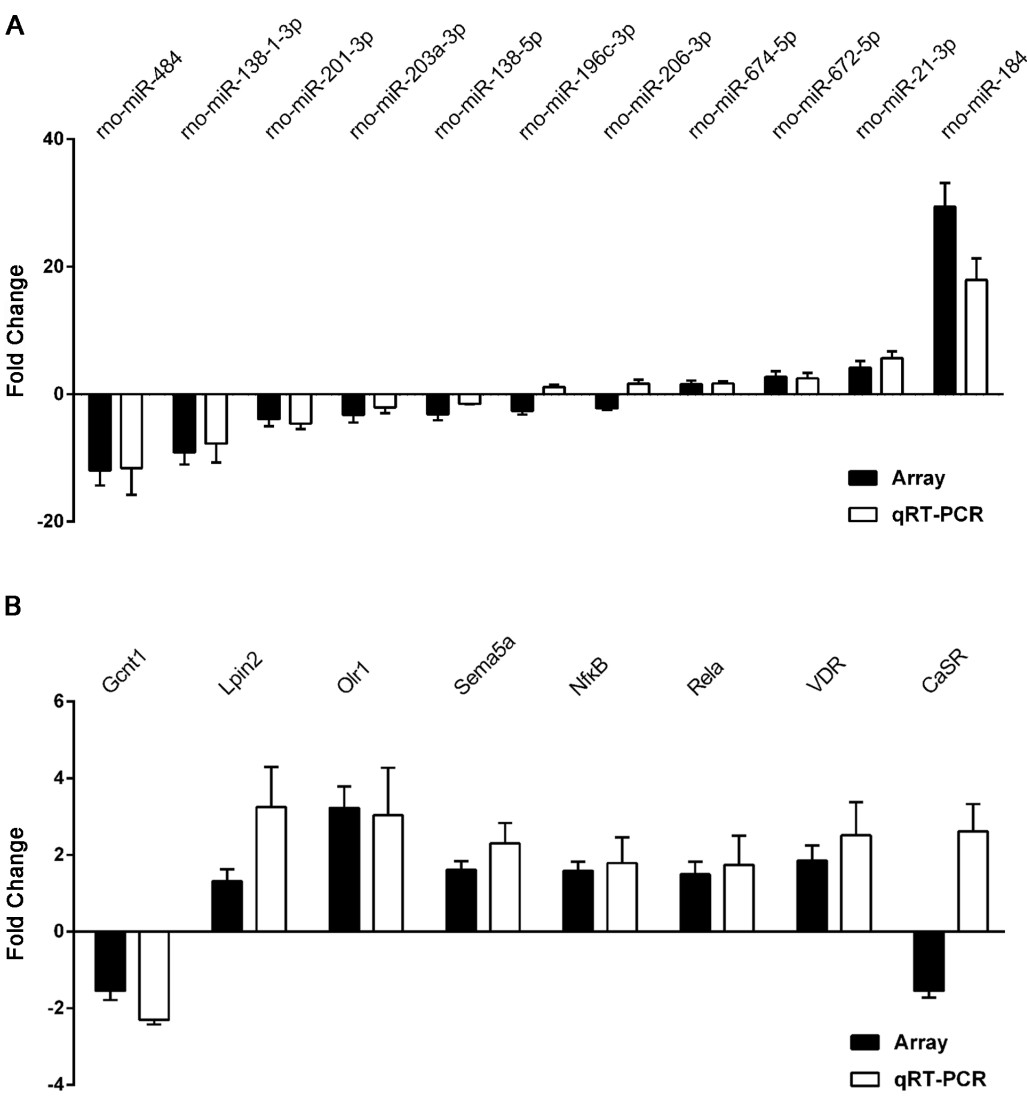

**Figure 6  qRT-PCR was performed to confirm the expression of 11 selected miRNAs and eight mRNAs.** Quantitative reverse transcription PCR (qRT-PCR) validation of differentially expressed miRNAs (A) and mRNAs (B). Results are shown as mean of log2(GHS/SD) and are presented side by side with the respective microarray results.

MicroRNAs regulate various disease processes by inhibiting the expression of their target mRNAs. Understanding the relevance of miRNA and mRNA expression patterns in GHS rat kidneys is important to better elucidate the relationship between pathophysiological process and genes. In the present study, we used bioinformatics methods to determine the potential role of differentially expressed miRNAs and mRNAs in GHS rat kidneys, and identified specific miRNAs and possible negative regulative mRNAs that may affect the development of hypercalciuria.

We used a multi-step approach to identify mRNA targets of dysregulated miRNAs in GHS rat kidneys whereby we first identified mRNAs ($N = 2,418$) and miRNAs ($N = 19$) that were significant differentially expressed between GHS and control SD rats; next, potential

miRNA-mRNA pairs were predicted for the 19 miRNAs using miRanda, resulting in 29,164 miRNA-mRNA pairs; the intersecting set of predicted target mRNAs and differentially expressed mRNAs were then selected for further analysis.

We found a significant enrichment in over 700 GO terms, including inflammatory response (GO:0006954) and response to hypoxia (GO:0001666). Interestingly, local hypoxic conditions and inflammatory injuries have been linked to the initiation of urolithiasis (*Cao et al., 2006*), whereas antioxidants protect renal tubular cells from cellular injury and decrease the formation of calcium oxalate stones (*Itoh et al., 2005*). Moreover, the GO term response to calcium ion (GO:0051592) was enriched (based on differentially expressed genes such as IL-6 and VDR). Increased levels of VDR protein in the kidney are regarded as a common feature in both GHS rats and IH individuals (*Frick, Krieger & Bushinsky, 2015*), and VDR intimately affects urolithiasis formation (*Arcidiacono et al., 2014*; *Zhang, Nie & Jiang, 2013*). *Glybochko et al. (2010)* found a significant increase in serum IL-6 in nephrolithiasis patients compared with healthy individuals, and *Hasna et al. (2015)* showed that IL-6 was significantly increased in patients with diabetes mellitus and urolithiasis compared with patients with diabetes mellitus alone. In addition, skeletal system development (GO:0001501) and skeletal system morphogenesis (GO:0048705) were significantly increased in GHS rats. Our previous studies demonstrated the occurrence of osteochondral differentiation progression in primary renal tubular epithelial cells in GHS rats (*He et al., 2015*), which may be linked to the pathogenesis of calcium stone development.

By identifying pathway membership of dysregulated mRNAs, we can improve our understanding of underlying disease-related processes. We detected 93 enriched pathways including the MAPK signaling pathway, mineral absorption as well as the TGF-beta signaling pathway. *Khan (2013)* reported that the p38 MAPK/JNK pathway regulates crystallization modulator production and influences plaque formation as well as calcium oxalate nephrolithiasis. Our previous research also showed that calcium and TGF-$\beta$ may participate in the pathogenesis of epithelial-mesenchymal transition and lead to stone formation (*He et al., 2015*).

To better understand the biological processes linked to differentially expressed miRNAs and their predicted target genes, we constructed an interaction network where rno-miR-674-5p, rno-miR-672-5p, rno-miR-138-5p and rno-miR-21-3p were found to be highly connected, which means that they may play crucial roles in the regulatory network. Interestingly, *Wang et al. (2011)* reported that miR-674-5p was able to stimulate the expression of osteogenic marker genes; miR-138-5p is a risk factor for pancreatic cancer (*Yu et al., 2015*) and Alzheimer's disease (*Lugli et al., 2015*); and miR-21-3p increases resistance to cisplatin in a range of ovarian cell lines (*Pink et al., 2015*). In addition, rno-miR-484, rno-miR-138-1-3p, rno-miR-201-3p and rno-miR-203a-3p are downregulated in the network, nevertheless, previous studies reported these miRNAs to be tumor-associated (*Pizzini et al., 2013*; *Liu et al., 2015*; *Merhautova et al., 2015*; *Murray et al., 2014*; *Yang et al., 2016*; *Ye et al., 2015*). To our knowledge, there has been very little research on the relationship between these miRNAs and urolithiasis or calcium metabolism. *Hu et al. (2014)* reported serum and urinary levels of miR-155 were significantly elevated in patients with nephrolithiasis, and

miR-155 might influence pathophysiology of nephrolithiasis via regulating inflammatory cytokines expression. However, the expressions of miR-155 were not significantly different between the two groups of our present study. Regarding mRNAs in the constructed network, Sema5a, which belongs to the semaphorin gene family, had the highest degree of connectivity of 13. It has previously been reported that Sema5a increases cell proliferation and metastasis and suppresses tumor formation (*Lu et al., 2010*; *Sadanandam et al., 2012*). In addition, *Ding et al. (2008)* suggested that Sema5a is a risk factor for Parkinson's disease. Lpin2, which is associated with fatty acid, triacylglycerol, and ketone body metabolism (*Chen et al., 2015*), also had a high degree of connectivity of 9. Olr1, a crucial regulator of lipid metabolism (*Tejedor et al., 2015*), had a degree of connectivity of 8. Increasing numbers of studies are reporting a strong correlation between obesity/dyslipidemia and kidney stones (*De, Liu & Monga, 2014*; *Fujimura et al., 2014*). However, the mechanism whereby obesity and kidney stone disease are linked is still unknown. Recent research has indicated that obesity may increase reactive oxygen species and oxidative stress, which would influence the interaction between calcium oxalate/calcium phosphate crystals and renal epithelial cells (*Khan, 2012*).

The NF-kappa B signaling pathway was also significantly enriched in GHS rats; in addition, signal-net analysis has shown that NF-kappa B family might be a core role of our gene regulatory network including the members of NF-kappa B1, NF-kappa B2 and RelA. NF-kappa B is a transcription regulator that is activated by various intra- and extra-cellular stimuli such as cytokines, oxidant-free radicals, and bacterial or viral products (*Hoesel & Schmid, 2013*). Activated NF-kappa B translocates into the nucleus and stimulates the expression of genes involved in a wide variety of biological functions. *Menon et al. (2014)* found that the receptor activator of NF-kappa B ligand mediates bone resorption in IH, while *Tozawa et al. (2008)* reported that oxalate induced OPN expression by activating NF-kappa B in renal tubular cells. But there has been little research focusing on the function of NF-kappa B in the formation of idiopathic hypercalciuria urolithiasis. Furthermore, abnormal fat metabolism would also activate the NF-kappa B signaling pathway (*Yang et al., 2015*), which might contribute to the increased risk of urolithiasis in obesity populations. However, the molecular mechanism of NF-kappa B in IH disease is in need of further exploration, which may assist in improving the clinical diagnosis and treatment of patients with IH.

## CONCLUSIONS

Here, we successfully identify rat hypercalciuria-related miRNAs and their target mRNAs. This comprehensive analysis will provide valuable insights for future research, which should aim at confirming the role of these genes in the pathophysiology of hypercalciuria stone desease. More further work needs to be done to get the whole picture right in the disease context. To the best of our knowledge, this is the first study to focus on the role of miRNA-mRNA interactions in hypercalciuria urolithiasis.

## ACKNOWLEDGEMENTS

We thank GMINIX Informatics Ltd. Co (Shanghai, PR China) for their technical assistance with bioinformatics analysis.

### Funding

This study was supported by the National Natural Science Foundation of China (No. 81270787). The funders had no role in study design, data collection and analysis, decision to publish, or preparation of the manuscript.

### Grant Disclosures

The following grant information was disclosed by the authors:
National Natural Science Foundation of China: 81270787.

### Competing Interests

The authors declare there are no competing interests.

### Author Contributions

- Yuchao Lu conceived and designed the experiments, analyzed the data, wrote the paper, prepared figures and/or tables, reviewed drafts of the paper.
- Baolong Qin conceived and designed the experiments, analyzed the data, prepared figures and/or tables, reviewed drafts of the paper.
- Henglong Hu analyzed the data, wrote the paper, reviewed drafts of the paper.
- Jiaqiao Zhang, Yufeng Wang and Qing Wang performed the experiments, reviewed drafts of the paper.
- Shaogang Wang conceived and designed the experiments, wrote the paper, reviewed drafts of the paper.

### Animal Ethics

The following information was supplied relating to ethical approvals (i.e., approving body and any reference numbers):

All animal procedures were approved by the Ethical Committee of Tongji Hospital, Tongji Medical College, Huazhong University of Science and Technology (No. TJ-A20141211).

### Microarray Data Deposition

The following information was supplied regarding the deposition of microarray data:
GEO Series accession number GSE75543.

### Supplemental Information

Supplemental information for this article can be found online at http://dx.doi.org/10.7717/peerj.1884#supplemental-information.

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
