# Peer review of "Integrative microRNA-gene expression network analysis in genetic hypercalciuric stone-forming rat kidney"

_PeerJ, doi:10.7717/peerj.1884_

## Round 0.1 · original submission · Major Revisions

The work done by Lu et al is a well-designed study but, I have a serious concern about the data analysis and its presentation.

The data is not interpreted clearly. For example, the conclusion drawn from table S2 must be included as a table in the manuscript for clear understanding.

The microarray-based analysis is not clearly written.

It is not shown the most significantly regulated miRNA-mRNA hybrid.
In the materials and methods section (line 87-89), RNA integrity can not be evaluated by Nanodrop or Agarose gel electrophoresis. Please clarify it.

Reviewer 1 ·

Basic reporting

This is a well-written and well-organized paper. The study reveals the role of miRNA-mRNA in the disease hypercalciuria urolithiasis and makes a valuable contribution to understand the disease better. This could pave way for future research aimed to improve disease diagnosis and treatments.
Page 3 Methods and Results, line 2 should read “...bioinformatics analysis was adopted to construct...” not conduct as written
Line 29, 30 of the manuscript (methods and results section) needs to be rephrased.
Lines 259, 260 grammar errors need to be fixed.
There are some minor typos in the manuscript that need to be fixed.

Experimental design

Experiments are relevant and rigorous.

Validity of the findings

Kindly elaborate the results obtained in the signal-net analysis (lines 169-171).
The study no doubt identifies rat hypercalciuria-related miRNAs and their target mRNAs but further work needs to be done to get the whole picture right in the disease context. The “we present a global view...” part should be rephrased.

·

Basic reporting

There are a few weaknesses in the manuscript, generally attributable to a perhaps too terse style. For example, in the Results lines 169-171, the authors state “Moreover, to investigate the regulatory relationships between these genes and their potential role in hypercalciuria, we performed a signal-net analysis based on significantly regulated KEGG pathways (Fig. 6, Table S12)” with no further reporting of the results in this section. I presume the darker bars in Fig. 4 indicate pathways of particular interest, but this is not indicated anywhere. In Fig. 7 Legend, panels A and B should be defined.

Experimental design

no comment

Validity of the findings

The object of this study was to determine if there is differential regulation of microRNAs (miRNA) in the kidney of genetic hypercalciuric stone-forming (GHS) rats compared to the founder strain, Sprague-Dawley (SD). 19 miRNAs and 2418 genes were found to be differentially expressed by microarray analysis. 11 miRNAs and 8 target RNAs were chosen for further analysis of expression and validated by qRT-PCR. Integrated pathway analysis indicated that NFB1 may play an important role in hypercalciuria and stone formation in GHS.

Additional comments

This is a well-designed study which arrives at interesting conclusions that may be relevant to hypercalciuric stone disease.

Reviewer 3 ·

Basic reporting

No comments

Experimental design

No comments

Validity of the findings

No comments

Additional comments

In this manuscript, the authors examined the correlation between differentially expressed miRs and mRNAs in a hypercalciuria rat model and applied transcriptomic approaches and bioinformatic tools for analysis of pahtways. While the experimental part is convincing, analysis and interpretation of the data have substantial problems. Therefore, I feel that this manuscript does not present findings novel enough to warrant publication in PeerJ.
Major points:
# In fig. 3, miR expression of three 'healthy' and three GHS rats were shown in a heat map. What is about the other nine animals analyzed?
# 19 miRs were identified as differentially expressed, however, only a set of 11 miRs were validated by quantitative RT-PCR. Why?
# The authors do not differentiate between direct target binding of miRs (formation of a miR : mRNA hybrid) and indirectly targeted mRNAs (subsequent regulation of mRNA turn-over by primarily controlled gene products).
# Several signaling/effector pathways were identified by in silico analysis (fig. 4), however, this is not remarked in the discussion section (e.g., what is the role of several virus-related pathways?).
# Fig. 5: What does it mean and what is it good for? Global analysis of cellular factors provides beyond doubt artificial signaling networks. It would be more instrumental to focus on literature data of the 19 miRs identified in this study.

---

## Round 0.2 · accepted · Accept

The manuscript has been improved, significantly. But, Language and the formatting problem still persist, and should be checked very carefully (e.g. see line 284).

Reviewer 1 ·

Basic reporting

The authors have addressed the rebuttal well and incorporated suggestions. The new manuscript have a few typos ( mainly spacing between words) that should be corrected before publishing.

Experimental design

No Comments

Validity of the findings

No Comments

Additional comments

It is an important piece of work and is well written.